# Random Positive-Only Projections:
# PPMI-Enabled Incremental Semantic Space Construction

## Abstract

We introduce *positive-only projection* (PoP), a novel technique for constructing semantic spaces and word embeddings. The PoP method is based on random projections. Hence, it is highly scalable and computationally efficient. In contrast to previous methods that use random projection matrices $\mathbf{R}$ with the expected value of 0 (i.e., $E(\mathbf{R}) = 0$), the proposed method uses $\mathbf{R}$ with $E(\mathbf{R}) > 0$. We use *Kendall*'s $\tau_b$ distance to compute vector similarities in the resulting non-Gaussian spaces. Most importantly, since $E(\mathbf{R}) > 0$, weighting methods such as positive pointwise mutual information (PPMI) can be applied to PoP-constructed spaces after their construction for efficiently transferring PoP embeddings onto spaces that are discriminative for semantic similarity assessments. Our PoP-constructed models, combined with PPMI, achieve an average score of 0.75 in the *MEN relatedness test*, which is comparable to results obtained by state-of-the-art top-performing algorithms.

## 1 Introduction

The development of data-driven methods of natural language processing starts with an educated guess, a distributional hypothesis: We assume that some properties of linguistic entities can be modelled by 'some statistical' observations in language data. In the second step, this statistical information (which is determined by the hypothesis) is collected and represented in a mathematical framework. In the third step, tools provided by the chosen mathematical framework are used to implement a similarity-based logic to identify linguistic structures, and/or to verify the proposed hypothesis. Harris's distributional hypothesis is a well-known example of step one that states that meanings of words correlate with the context in which the words appear. Vector space models and $\eta$-normed-based similarity measures are renowned examples for steps two and three, respectively (i.e., word space models or word embeddings).

However, as pointed out for instance by Baroni et al. (2014), the *count-based models* resulting from the steps two and three are not discriminative enough to achieve satisfactory results; instead, *predictive models* are required. To this end, an additional transformation step is often added. Turney and Pantel (2010) describe this additional step as a combination of *weighting* and *dimensionality reduction*.[1] This transformation from count-based to predictive models can be implemented simply via a collection of rules of thumb (i.e., heuristics), and/or it can involve more sophisticated mathematical transformations, such as converting raw counts to probabilities and using matrix factorization techniques. Likewise, by exploiting large amounts of computational power available nowadays, this transformation can be achieved via neural word embedding techniques (Mikolov et al., 2013; Levy and Goldberg, 2014).

To a large extend, the need for such transformations arises from the *heavy-tailed* distributions that we often find in statistical natural language models (such as the Zipfian distribution of words in contexts when building word spaces). Consequently, count-based models are sparse and high-dimensional and therefore both computationally expensive to manipulate (due of the high dimensionality of models) and nondiscriminatory (due to the combination of the high-dimensionality of the models and the sparseness of observations, see Minsky and Papert (1969, chap. 12)).

---

[1] Similar to topics of feature weighting, selection, and engineering in statistical machine learning.

On the one hand, although neural networks are the top performers for addressing this problem, their usage is highly costly: they need to be trained that is often very time-consuming,[2] and their performance can vary from one task to another depending on their *objective function*.[3] On the other hand, although methods based on random projections efficiently address the problem of reducing the dimensionality of vectors—such as RI (Kanerva et al., 2000), reflective random indexing (RRI), (Cohen et al., 2010), ISA (Baroni et al., 2007) and random Manhattan indexing (RMI) (Zadeh and Handschuh, 2014)—in effect they retain distances between entities in the original space.[4] In addition, since these methods use asymptotic Gaussian or Cauchy random projection matrices $\mathbf{R}$ with $\mathrm{E}(\mathbf{R}) = 0$, their resulting vectors cannot be adjusted and transformed using weighting techniques such as PPMI. Hence these methods do not outperform neural embeddings and combinations of PPMI weighting of count-baseds model followed by matrix factorization—such as truncation of weighted-vectors using singular value decomposition (SVD).

In order to overcome these problems, we propose a new method called *positive-only projection (PoP)*. PoP is an incremental semantic space construction method that is similar to RI in the sense that it employs random projections. Hence, the construction of models using PoP does not require prior computation of embeddings but simply generating random vectors. However, in contrast to RI and similar methods, the PoP-constructed spaces can undergo weighting transformations such as PPMI. This is due to the fact that PoP uses random vectors that contain only positive integer values. Since the method is based on random projections, models can be built incrementally and efficiently. Since the vectors in PoP-constructed models are small (i.e., dimensionality of a few hundred), applying weighting methods such as PPMI to these models is incredibly faster than applying them to classical count-based models. Combined with a suitable weighting method such PPMI, the PoP method yields competitive results concern-

---

[2]Baroni et al. (2014) state that it took *Ronan Collobert* two months to train a set of embeddings from a Wikipedia dump. Even using GPU-accelerated computing, the required computation and training time for inducing neural word embeddings is high.

[3]Ibid, see results reported in supplemental materials.

[4]For $\eta$-normed space that they are designed for, i.e., $\eta = 2$ for RI, RRI, and ISA and $\eta = 1$ for RMI.

ing accuracy in semantic similarity assessment, compared for instance to neural net-based approaches and combinations of count-based models with weighting and matrix factorization. These results, however, are achieved without the need for heavy computations. Thus, instead of hours, days or months, models can be built in a matter of a few seconds or minutes. Note that even without weighting transformation, PoP-constructed models display a better performance than RI on tasks of semantic similarity assessments.

We describe the PoP method in § 2. In order to evaluate our models, in § 3, we report the performance of the PoP method in the MEN relatedness test. Finally, § 4 concludes with a discussion.

## 2 Method

### 2.1 PoP-based Model Construction

Any transformation from a count-based model to a predictive one can be expressed using a matrix notation such as:

$$\mathbf{C}_{p \times n} \times \mathbf{T}_{n \times x} = \mathbf{P}_{p \times x}. \qquad (1)$$

In Equation 1, $\mathbf{C}$ denotes the count-based model consisting of $p$ vectors and $n$ context elements (i.e., $n$ dimensions). $\mathbf{T}$ is the transformation matrix that maps the $p$ $n$-dimensional vectors in $\mathbf{C}$ to an $x$-dimensional space (often, but not necessarily, $x \neq n$ and $x \ll n$). Finally, $\mathbf{P}$ is the resulting $x$-dimensional predictive model. Note that $\mathbf{T}$ can be a composition of several transformations, e.g., a weighting transformation $\mathbf{W}$ followed by a projection onto a space of lower dimensionality $\mathbf{R}$, i.e., $\mathbf{T}_{n \times x} = \mathbf{W}_{n \times n} \times \mathbf{R}_{n \times x}$.

In the proposed PoP technique, the transformation $\mathbf{T}_{n \times m}$ (for $m \ll n$, e.g., $100 \leq m \leq 5000$) is simply a randomly generated matrix. The elements $t_{ij}$ of $\mathbf{T}_{n \times m}$ have the following distribution:

$$t_{ij} = \begin{cases} 0 & \text{with probability } 1 - s \\ \lfloor \frac{1}{U^\alpha} \rfloor & \text{with probability } s \end{cases}, \qquad (2)$$

in which U is an independent uniform random variable in $(0, 1]$, $s$ is an extremely small number (e.g., $s = 0.01$) such that each row vector of $\mathbf{T}$ has at least one element that is not 0 (i.e., $\sum_{i=1}^{m} t_{ji} \neq 0$ for each row vector $t_j \in \mathbf{T}$). For $\alpha$, we choose $\alpha = 0.5$. Given Equations 1 and 2 and using the distributive property of multiplication over addition in matrices,[5] the desired semantic space (i.e., $\mathbf{P}$ in Equation 1) can be constructed

---

[5]That is $(\mathbf{A} + \mathbf{B}) \times \mathbf{C} = \mathbf{A} \times \mathbf{C} + \mathbf{B} \times \mathbf{C}$.

using the two-step procedure of incremental word space construction known from RI:

**Step 1.** Each context element is mapped to one $m$-dimensional *index vector* $\vec{r}$. $\vec{r}$ is randomly generated such that most elements in $\vec{r}$ are 0 and only a few are positive integers (i.e., the elements of $\vec{r}$ have the distribution given in Equation 2).

**Step 2.** Each target entity that is being analysed in the model is represented by a *context vector* $\vec{v}$ in which all the elements are initially set to 0. For each encountered occurrence of this target entity together with a context element (e.g., through a sequential scan of a corpus), we update $\vec{v}$ by adding the index vector $\vec{r}$ of the context element to it.

This process results in a model built directly at the reduced dimensionality $m$ (i.e., $\mathbf{P}$ in Equation 1). The first step corresponds to the construction of the randomly generated transformation matrix $\mathbf{T}$: Each index vector is a row of the transformation matrix $\mathbf{T}$. The second step is an implementation of the matrix multiplication in Equation 1 which is distributed over addition: Each context vector is a row of $\mathbf{P}$, which is computed in an iterative process.

## 2.2 Measuring Similarity

Once $\mathbf{P}$ is constructed, if desirable, similarities between entities can be computed by their *Kendall*'s $\tau_b$ ($-1 \leq \tau_b \leq 1$) correlation (Kendall, 1938). In order to compute $\tau_b$, we need to define a number of values. Given vectors $\vec{x}$ and $\vec{y}$ of the same dimension, we call a pair of observations $(x_j, y_j)$ and $(x_{j+1}, y_{j+1})$ in $\vec{x}$ and $\vec{y}$ *concordant* if $(x_j < x_{j+1} \wedge y_j < y_{j+1}) \vee (x_j > x_{j+1} \wedge y_j > y_{j+1})$. The pair is called *discordant* if $(x_j < x_{j+1} \wedge y_j > y_{j+1}) \vee (x_j > x_{j+1} \wedge y_j < y_{j+1})$. Finally, the pair is called *tied* if $x_j = x_{j+1} \vee y_j = y_{j+1}$. Note that a tied pair is neither concordant nor discordant. We define $n_1$ and $n_2$ as the number of pairs with tied values in $\vec{x}$ and $\vec{y}$, respectively. We use $n_c$ and $n_d$ to denote the number of concordant and discordant pairs, respectively. If $m$ is the dimension of the two vectors, then $n_0$ is defined as the total number of observation pairs: $n_0 = \frac{m(m-1)}{2}$. Given these definitions, Kendall's $\tau_b$ is given by

$$\tau_b = \frac{n_c - n_d}{\sqrt{(n_0 - n_1)(n_0 - n_2)}}.$$

To compute $\tau_b$, we adopt an implementation of the algorithm proposed by Knight (1966), which has a computational complexity of $O(n \log n)$.[6] Since the vectors resulting from the PoP method have a very low dimensionality, this complexity does not harm the overall efficiency of the approach.

The choice of $\tau_b$ is motivated by generalising the role that cosine plays for computing similarities between vectors that are derived from a standard Gaussian random projection. In random projections with $\mathbf{R}$ of (asymptotic) $\mathcal{N}(0,1)$ distribution, despite the common interpretation of the cosine similarity as the angle between two vectors, cosine can be seen as a measure of product-moment correlation coefficient between the two vectors. Since $\mathbf{R}$ and thus the obtained projected spaces have $E = (0)$, Pearson's correlation and the cosine measure have the same definition in these spaces (see also Jones and Furnas (1987) for a similar claim and on the relationships between correlation and the inner product and cosine). Subsequently, one can propose that in Gaussian random projections, Pearson's correlation is used to compute similarities between vectors.

However, the use of projections proposed in this paper (i.e., $\mathbf{T}$ with a distribution set in Equation 2) will result in vectors that have a non-Gaussian distribution. In this case, $\tau_b$ becomes a reasonable candidate for measuring similarities (i.e., correlations between vectors) since it is a nonparametric correlation coefficient measure that does not assume a Gaussian distribution of projected spaces. However, we do not exclude the use of other similarity measures and may apply them in future work. In particular, we envisage additional transformations of PoP-constructed spaces to induce vectors with Gaussian distributions (see for instance the log-based PPMI transformation used in the next section). If a transformation to a Gaussian-like distribution is performed, then the use of Pearson's correlation, which works under the assumption of Gaussian distribution, yields better results than Kendall's correlation (as confirmed by our experiments).

## 2.3 Some Delineation of the PoP Method

The PoP method is a *randomized algorithm*. In this class of algorithms, at the expense of a tolerable loss in accuracy of the outcome of the computations (of course, with a certain acceptable amount of probability) and by the help of *ran-*

---

[6]In our evaluation, we use the implementation of Knight's algorithm in the *Apache Commons Mathematics Library*.

*dom decisions*, the computational complexity of algorithms for solving a problem is reduced (see, e.g., Karp (1991), for an introduction to randomized algorithms).[7] For instance, using Gaussian-based sparse random projections in RI, the computation of eigenvectors (often of complexity of $O(n^2 \log m)$) is replaced by a much simpler process of random matrix construction (of an approximate complexity of O(n))—see (Bingham and Mannila, 2001). In return, a randomized algorithm such as the RI and PoP methods give different results even for the same input.

Assume the difference between the optimum result and the result from a randomized algorithm is given by $\delta$ (i.e., the error caused by replacing deterministic decisions with random ones). Much research in theoretical computer science and applied statistics focuses on specifying bounds for $\delta$, which is often expressed as a function of the probability $\epsilon$ of encountered errors. For instance, $\delta$ and $\epsilon$ in Gaussian random projections are often derived from the lemma proposed by Johnson and Lindenstrauss (1984) and its variations. Similar studies for random projections in $\ell_1$-normed spaces and deep neural networks are Indyk (2000) and Arora et al. (2014), respectively.

At this moment, unfortunately, we are not able to provide a detailed mathematical account for specifying $\delta$ and $\epsilon$ for the results obtained by the PoP method (nor are we able to pinpoint theoretical discussion about PoP's underlying random projection). Instead, we rely on the outcome of our simulations and the performance of the method in an NLP task. Note that this is not an unusual situation. For instance, Kanerva et al. (2000) proposed RI with no mathematical justification. In fact, it was only a few years later that Li et al. (2006) proposed mathematical lemmas for justifying very sparse Gaussian random projections such as RI (QasemiZadeh, 2015). At any rate, projection onto manifolds is a vibrant research both in theoretical computer science and in mathematical statistics. Our research will benefit from this in the near future. Concerning $\delta$, it can be shown that it and its variance $\sigma_\delta^2$ are functions of the dimension $m$ of the projected space, that is: $\sigma_\delta^2 \approx \frac{1}{m}$, based on similar mathematical principles proposed by Kaski (1998) (and of Hecht-Nielsen (1994)) for random mapping methods.

Our empirical research and observations on language data show that projections using the PoP method exhibit similar behavioural patterns as other sparse random projections in $\alpha$-normed spaces. The dimension $m$ of random index vectors can be seen as the capacity of the method to memorize and distinguish entities. For $m$ up to a certain number ($100 \le m \le 4000$) in our experiments, as was expected, a PoP-constructed model for a large $m$ shows a better performance and smaller $\delta$ than a model for a small $m$. Since observations in semantic spaces have a *very*-long-tailed distribution, choosing different values of non-zero elements for index vectors does not effect the performance (as mentioned, in most cases 2 or 3 non-zero elements are sufficient). Furthermore, changes in the adopted distribution of $t_{ij}$ only slightly effect the performance of the system, due to the use of $\tau_b$ as a similarity measure.

In the next section, using empirical investigations we show the advantages of the PoP model and support the claims from this section.

## 3 Evaluation & Empirical Investigations

### 3.1 Comparing PoP and RI

For evaluation purposes, we use the MEN relatedness test set (Bruni et al., 2014) and the UKWaC corpus (Baroni et al., 2009). The dataset consists of 3000 pairs of words (from 751 distinct tagged lemmas). Similar to other 'relatedness tests', Spearman's rank correlation $\rho$ score from the comparison of human-based ranking and system-induced rankings is the figure of merit. We use these resources for evaluation since they are in public domain, both the dataset and corpus are large, and they have been used for evaluating several word space models—for example, see Levy et al. (2015), Tsvetkov et al. (2015), Baroni et al. (2014), Kiela and Clark (2014). In this section, unless otherwise stated, we use cosine for similarity measurements.

Figure 1 shows the performance of the simple count-based word space model for lemmatized-context-windows that extend symmetrically around lemmas from MEN.[8] As expected, up to a certain context-window size, the performance using count-based methods increases with an

---

[7] Such as many classic search algorithms that are proposed for solving NP-complete problems in artificial intelligence.

[8] We use the tokenized preprocessed UKWaC. However, except for using part-of-speech tags for locating lemmas listed in MEN, we do not use any additional information or processes (i.e., no frequency cut-off for context selection, no syntactic information, etc.).

extension of the window.[9] For context-windows larger than 25+25 the performance gradually declines. More importantly, in all cases we have $\rho < 0.50$.

We performed the same experiments using the RI technique. For each context window size, we performed 10 runs of the RI model construction. Figure 1 reports for each context-window size the average of the observed performances for the 10 RI models. In this experiment, we used index vectors of dimensionality 1000 containing 4 non-zero elements. As shown in Figure 1, the average performance of the RI is almost identical to the performance of the count-based model. This is an expected result since RI's objective is to retain Euclidean distances between vectors (thus cosine) but in spaces of lowered dimensionality. In this sense, RI is successful and achieves its goal of lowering the dimensionality while keeping distances between vectors. But it does not yield any improvements in the similarity assessment task.

We then performed similar experiments using PoP-constructed models, with the same context window sizes and the same dimensions as in the RI experiments, averaging again over 10 runs for each context window size. The performance is also reported in Figure 1. For the PoP method, however, instead of using the cosine measure we use $\tau_b$ for measuring similarity. The PoP-constructed models converge faster than RI and count-based method and for smaller context-windows they outperform the count-based and RI methods with a large margin. However, as the size of the windows grow, performances of these methods become more similar (but PoP still outperforms the others). One possible interpretation is that PoP is more sensitive to noise; if that is the case, this can be addressed by increasing the dimensionality of index vectors in order to reduce distortions in the projected spaces. In any case, the performance of PoP remains above 0.50 (i.e., $\rho > 0.50$).

### 3.2 PPMI Transformation of PoP Vectors

Although PoP outperforms RI and count-based models, its performance is still not satisfying since its index vectors are not sufficiently discriminative. In order to remedy this, transformations based on association measures such as positive pointwise mutual information (PPMI) have been

---

[9]After all, in models for relatedness tests, relationships of topical nature play a more important role than other relationships such as synonymy.

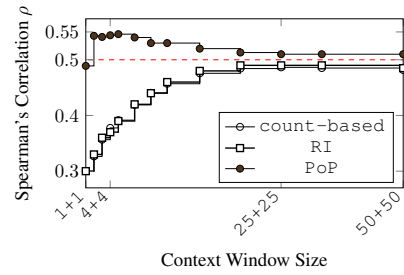

Figure 1: Performance of the classic count-based a-word-per-dimension model vs. RI vs. Pop.

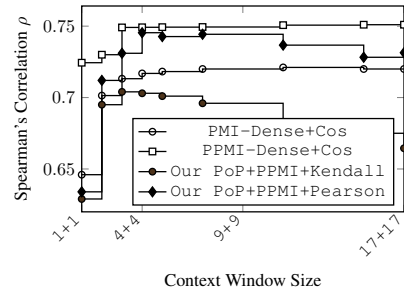

Figure 2: Performances of (P)PMI-transformed models for various sizes of context-windows. From context size 4+4, the performance remains almost intact (0.72 for PMI and 0.75 for PPMI). We also report the average performance for PoP-constructed models constructed at the dimensionality $m = 1000$ and $s = 0.002$. PoP+PPMI+Pearson exhibits a performance similar as dense PPMI-weighted models, however, much faster and using far less amount of computational resources. Note that reported PoP+PMI performances can be enhanced by using $m > 1000$.

proposed for the adjustment of weights (which correspond to coordinates of vectors). This often enhances the discriminatory power of the models and thus increases their performance in semantic similarity assessment tasks (see Church and Hanks (1990), Turney (2001), Turney (2008), and Levy et al. (2015)). For a given set of vectors, PMI is interpreted as a measure of information overlap between vectors. As put by Bouma (2009), PMI is a mathematical tool for measuring how much the actual probability of a particular co-occurrence (e.g., two words in a word space) deviate from the expected probability of their individual occurrences (e.g., the probability of occurrences of each word in a words space) under the assumption of independence (i.e., the occurrence of one word does not affect the occurrences of other words).

In Figure 2, we show the performance of PMI-

transformed spaces for small context-windows. One major problem with PMI is that it requires a lot of resources for its computation (as put by, PMI-induced spaces are dense). Even for a small number of entities in the model, due to the power-law distribution of word co-occurrences, the resulting spaces from large context-windows are high-dimensional so that the computation of PMI weights becomes intractable. But still, even for small context-windows, PMI+Cosine models outperform other techniques including the introduced count-based PoP method. The performance of PMI models can be further enhanced by its normalization, often discarding negative values[10] and using Positive PMI values (PPMI). Also, SVD truncation of PPMI-weighted spaces can improve the performance slightly (see the above mentioned references) requiring, however, expensive computations of eigenvectors.[11] For a $p \times n$ matrix with elements $v_{xy}$, $1 \leq x \leq p$ and $1 \leq y \leq n$, we compute the PPMI weight for a component $v_{xy}$ as follows:

$$ppmi(v_{xy}) = \max(0, \log \frac{v_{xy} \times \sum_{i=1}^{p} \sum_{j=1}^{n} v_{ij}}{\sum_{i=1}^{p} v_{iy} \times \sum_{j=1}^{n} v_{xj}}). \quad (3)$$

The most important benefit of the PoP method is that PoP-constructed models, in contrast to previously suggested random projection-based models, can be still weighted using PPMI (or any other weighting techniques applicable to the original count-based models). In an RI-constructed model, the sum of values of row and column vectors of the model are always 0 (i.e., $\sum_{i=1}^{p} v_{iy}$ and $\sum_{j=1}^{n} v_{xj}$ in Equation 3 are always 0). As mentioned earlier, this is due to the fact that a random projection matrix in RI has an asymptotic standard Gaussian distribution (i.e., transformation matrix $\mathbf{R}$ has $E(\mathbf{R}) = 0$). As a result, PPMI weights for the RI-induced vector elements are undefined. In contrast to RI, the sum of values of vector elements in the PoP-constructed models is always greater than 0 (since the transformation is carried out by a projection matrix $\mathbf{R}$ of $E(\mathbf{R}) > 0$). Also, depending on the structure of data in the underlying count-based model, by choosing a suitably large value of $s$, it can be guaranteed that the sum of column vectors is always a non-zero value. Hence, vectors in PoP models can undergo the PPMI transformation defined in Equation 3. Moreover, the PPMI

transformation is much faster, compared to the one performed on count-based models, due to the low dimensionality of vectors in the PoP-constructed model. Therefore, the PoP method makes it possible to benefit both from the high efficiency of randomized techniques as well as from the high accuracy of PPMI transformation in semantic similarity tasks.

If we put aside the information-theoretic interpretation of PPMI weighting (i.e., distilling statistical information that really matters), the logarithmic transformation of probabilities in the PPMI definition plays the role of a *power transformation* process for converting long-tailed distributions in count-based models to Gaussian-like distributions in predictive models. From a statistical perspective, any variation of PMI transformation can be seen as an attempt to stabilize variance of vector coordinates and therefore to make the observations more similar/fit to Gaussian distribution (a practice with long history in many research, particularly in biological and psychological sciences).

To exemplify this phenomenon, in Figure 3, we show histograms of the distributions of the assigned weights to the vector that represents the lemmatized form of the verb 'abandon' in various models. As shown, the raw collected frequencies in the count-based model have a long tail distribution (see Figure 3a). Applying the log transformation to this vector yields a vector of weights with a Gaussian distribution (Figure 3b). Weights in the RI-constructed vector (Figure 3c) have a perfect Gaussian distribution but with expected value of 0 (i.e., $\mathcal{N}(0, 1)$). The PoP method, however, largely preserves the long tail distribution of coordinates from the original space (Figure 3d), which in turn can be weighted using PPMI and thereby transformed into a Gaussian-like distribution.

Given that models after PPMI transformation have bell-shaped Gaussian distributions, we expect that a correlation measure such as Pearson's $r$, which takes advantage of the prior knowledge about the distribution of data, outperforms the non-parametric Kendall's $\tau_b$ for computing similarities in PPMI-transformed spaces.[12] This is indeed the case (see Figure 2).

---

[10]See Bouma (2009) for a mathematical delineation. Jurafsky and Martin (2015) also provide an intuitive description.
[11]In our experiments, applying SVD truncation to models results in negligible improvements between 0.01 and 0.001.

[12]Note that using correlation measures such as Pearson's $r$ and Kendall's $\tau_b$ in count-based model may excel measures such as cosine. However, their application is limited due to the high-dimensionality of count-based methods.

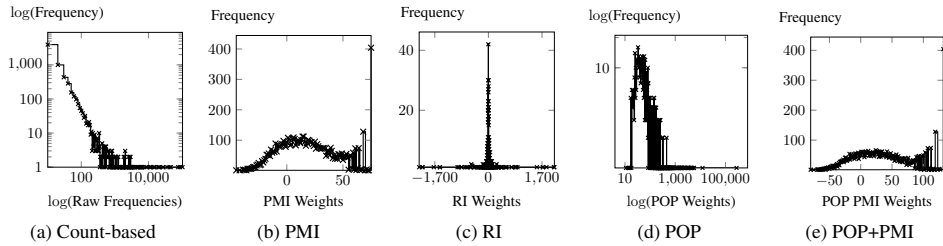

(a) Count-based (b) PMI (c) RI (d) POP (e) POP+PMI

Figure 3: A histogram of the distribution of frequencies of weights in various models from 1+1 context-windows for the lemmatized form of the verb 'abandon' in the UKWaC corpus.

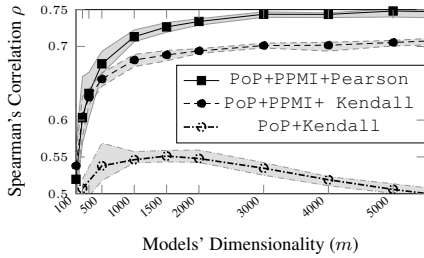

Figure 4: Changes in PoP's performance when the dimensionality of models increases. The average performance in each set-up is shown by marked lines. The margins around these lines show the minimum and maximum performance observed in 10 independent executions.

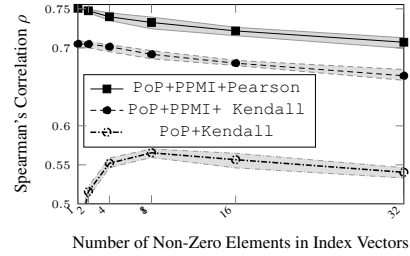

Figure 5: Changes in PoP's performances when the dimensionality of models are fixed to $m = 3000$ and the number of non-zero elements in index vectors (i.e., $s$) increases. The average performances in each set-up are shown by marked lines. The margins around these lines show the minimum and maximum performance observed in 10 independent executions.

### 3.3 PoP's Parameters and its Random Behaviour and Performance

As discussed in § 2.3, PoP is a randomized algorithm and its performance is influenced by a number of parameters. In this section, we study the PoP method's behaviour by reporting its performance in the MEN relatedness test under different parameter settings. To keep evaluations and reports in a manageable size, we focus on models built using context-windows of size 4+4.

Figure 4 shows the method's performance when the dimension $m$ of the projected index vectors increases. In these experiments, index vectors are built using 4 non-zero elements; thus, as $m$ increases, $s$ in Equation 2 decreases. For each $m$, $100 \leq m \leq 5000$, the models are built 10 times and the average as well as the maximum and the minimum observed performances in these experiments are reported. For PPMI transformed PoP spaces, with increasing dimensions, the performance boosts and, furthermore, the variance in performance (i.e., the shaded areas)[13] gets smaller.

However, for the count-based PoP method without PPMI transformation (shown by dash-dotted lines) and with the number of non-zero elements fixed to 4, increasing $m$ over 2000 decreases the performance. This is unexpected since an increase in dimensionality is usually assumed to entail an increase in performance. This behaviour, however, can be the result of using a very small $s$; simply put, the number of non-zero elements are not sufficient to build projected spaces with adequate distribution. To investigate this matter, we study the performance of the method with the dimension $m$ fixed to 3000 but with index vectors built using different numbers of non-zero elements, i.e., different values of $s$.

Figure 5 shows the observed performances. For PPMI-weighted spaces, increasing the number of non-zero elements clearly deteriorates the performance. For unweighted PoP models, an increase in $s$ up to the limit that does not result in non-orthogonal index vectors enhances performances. As shown in Figure 6, when the dimensionality

---

[13]Evidently, the probability of worst and best performances can be inferred from the reported average results.

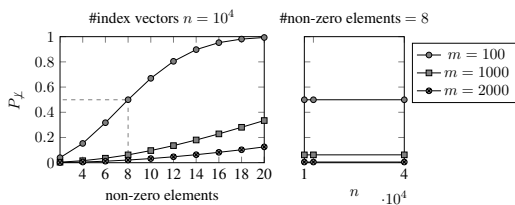

Figure 6: Proportion of non-orthogonal pairs of index vectors (i.e., $P_{\not\perp}$) obtained in a simulation for various dimensionality and number of non-zero elements. The left figure shows the changes of $P_{\not\perp}$ For a fixed set of index vectors $n = 10^4$ when number of non-zero elements increases. The right figure shows $P_{\not\perp}$ when the number of non-zero elements is fixed to 8 but the number of index vectors $n$ increases. As shown, $P_{\not\perp}$ is determined by the number of non-zero elements and dimensionality of index vectors and independently of $n$.

of the index vectors is fixed and $s$ increases, the chances of having non-orthogonal vectors in index vectors boosts. Hence, the chances of distortions in similarities increase. These distortions can enhance the result if they are controlled (e.g., using a training procedure such as the one used in neural net embedding). However, when left to chance, they can often lower the performance. Evidently, this is a simplified justification: in fact $s$ plays the role of a switch that controls resemblance between distribution of data in the original space and the projected/transformed spaces. It seems that the sparsity of vectors in the original matrix plays a role for finding the optimal value for $s$. If PoP-constructed models are used directly (together with $\tau_b$) for computing similarities, then we propose $0.002 < s$. If PoP-constructed models are subject to an additional weighting process for stabilising vector distributions into Gaussian-like distributions such as PPMI, we propose using only 1 or 2 non-zero elements.

Last but not least, we confirm that by carefully selecting context elements (i.e., removing stop words and using lower and upper bound frequency cut-offs for context selection) and fine tuning PoP+PPMI+Pearson (i.e., increasing the dimension of models and scaling PMI weights as in Levy et al. (2015)) we achieve an even higher score in the MEN test (i.e., an average of 0.78 with the max of 0.787). Moreover, although improvements from applying SVD truncation are negligible, we can employ it for reducing the dimensionality of PoP vectors (e.g., from 6000 to 200).

## 4 Discussion

We introduced a new technique called PoP for incremental construction of semantic spaces. PoP can be seen as a dimensionality reduction method, which is based on a newly devised random projection matrix $\mathbf{R}$ of $E(\mathbf{R})$. The major benefit of PoP is that it transfers vectors onto spaces of lower dimensionality without changing their distribution to a Gaussian shape with $E = 0$. Transformed spaces obtained using PoP can therefore be manipulated similarly to count-based models, only much faster and consequently requiring a considerably lower amount of computational resources.

PPMI weighting can be easily applied to POP-constructed models. In our experiments, we observe that PoP+PPMI+Pearson can be used to build models that achieve a high performance in semantic relatedness tests. More concretely, for index vector dimensions $m \geq 3000$, PoP+PPMI+Pearson achieves an average score of 0.75 in the MEN relatedness test, which is comparable to many neural embedding techniques (see scores reported in Chen and de Melo (2015) and Tsvetkov et al. (2015)). However, in contrast to these approaches, PoP+PPMI+Pearson achieves this competitive performance without the need for time-consuming training of neural nets. Moreover, the involved processes are all done on vectors of low dimensionality. Hence, the PoP method can dramatically enhance the performance of a number of distributional natural language analyses.

This research can be extended in several ways. Firstly, theoretical accounts for PoP still need to be investigated. Secondly, many new methods can be designed by combining the proposed PoP and serialization of various weighting techniques (such as previously done for RI). Moreover, since weighting processes can be applied at a very low dimensionality, many computationally-intense techniques which cannot be applied to high-dimensional models for stabilising distributions of vectors (e.g., the use of power transforms such as BoxCox that maximizes the correlation coefficient of a Gaussian distribution of the weighted vectors) are now available to be used. On a similar basis, in the process of developing neural embeddings, PoP-constructed models can replace the high-dimensional count-based models as an input. Lastly, an important avenue of research is the use of so-called derandomization techniques to make PoP models linguistically more informed.

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

## A Supplemental Material

If the paper accepted, codes and resulting embeddings from experiments will be shared alongside the camera ready version.

