# Peer review of "Random Positive-Only Projections: PPMI-Enabled Incremental Semantic Space Construction"

_CoNLL 2016 — decision unknown_

[Official Review · Reviewer 1 · rating 2 · confidence 4]
soundness 3 · originality 3 · clarity 3 · impact 2 · substance 3 · appropriateness 5 · meaningful comparison 2 · replicability 3 · presentation format Poster

I am buying some of the motivation: the proposed method is much faster to train
than it is to train a neural network. Also, it keeps some properties of the
distribution when going to lower dimensionality. 

However, I am not convinced why it is so important for vectors to be
transformable with PPMI.

Most importantly, there is no direct comparison to related work.

Detailed comments:

- p.3: The definition of Kendall's tau that the authors use is strange. This is
NOT the original formula; I am not sure what it is and where it comes from.

- p.3: Why not use Spearman correlation as is standard in semantic tasks (and
as teh authors do at evaluation time)?

- The datasets chosen for evaluation are not the standard ones for measuring
semantic relatedness that the NLP community prefers. It is nice to try other
sets, but I would recommend to also include results on the standard ones.

- I can only see two lines on Figure 1. Where is the third line?

- There is no direct comparison to related work, just a statement that 

Some typos:

- large extend -- extent

[Official Review · Reviewer 2 · rating 2 · confidence 3]
soundness 3 · originality 4 · clarity 2 · impact 3 · substance 4 · appropriateness 5 · meaningful comparison 4 · replicability 3 · presentation format Oral Presentation

The paper presents a positive-only projection (PoP) word embedding method. This
is a random projection method with a random projection matrix whose expected
value is positive. The authors argue that this enables the application of PPMI
which is not possible with an expected value of 0 and that being a random
projection method, their computation is efficient.

My main reservation about this paper has to do with its clarity. Particularly:

1. I could not understand the core difference between the method proposed in
the paper and previous random projection methods. Hence, I could not understand
how (and whether) the advantages the authors argue to achieve hold.

2. It was hard to follow the arguments of the paper starting from the
introduction. 

3. Some of the arguments of the paper are not supported: 

- Line 114: Sentence starts with "in addition"

- Line 137: Sentence starts with "Since"

- Line 154: Sentence starts with "thus"

4. While I have worked on vector space modeling (who hasn't ?), I am not an
expert to random projections and have not used them in my research. It was hard
for me to understand the logic behind this research avenue from the paper. I
believe that a paper should be self contained and possible to follow by people
with some experience in the field.

5. The paper has lots of English mistakes (86: "To a large extend", 142: "such
PPMI").

In addition, I cannot see why the paper is evaluating only on MEN. There are a
couple of standard benchmarks (MEN, WordSeim, SimLex and a couple of others) -
if you present a new method, I feel that it is insufficient to evaluate only on
one dataset unless you provide a good justification.

I recommend that the authors will substantially improve the presentation in the
paper and will resubmit to another conference.